# Durum Wheat Fresh Pasta Fortification with Trub, a Beer Industry By-Product

**DOI:** 10.3390/foods11162496

**Published:** 2022-08-18

**Authors:** Elisabetta Lomuscio, Federico Bianchi, Mariasole Cervini, Gianluca Giuberti, Barbara Simonato, Corrado Rizzi

**Affiliations:** 1Department of Biotechnology, University of Verona, Strada Le Grazie 15, 37134 Verona, Italy; 2Department for Sustainable Food Process, Università Cattolica del Sacro Cuore, Via Emilia Parmense 84, 29122 Piacenza, Italy

**Keywords:** brewing by-product, debittered trub, pasta fortification, in vitro protein digestion, starch hydrolysis, sensory analyses

## Abstract

Trub is a brewing by-product rich in proteins and fibers. We used trub, after a debittering step, at 5, 10, and 15 g/100 g (PT5, PT10, and PT15, respectively) to fortify durum wheat fresh pasta. Technological and physical–chemical properties, in vitro digestibility, and sensorial characteristics of fortified pasta were determined. The technological aspects of the products were peculiar, suggesting the existence of complex interactions between the gluten network and starch with debittered trub powder. The fortified pasta samples showed a lower glucose release than the control at the end of in vitro starch hydrolysis. Furthermore, in vitro protein digestion rose only in PT15. PT5 and PT10 samples overcame the sensory acceptability threshold of 5, while PT15 showed the lowest acceptability. Debittered trub represents a suitable ingredient in fortified fresh pasta formulation with an up to 10% substitution level without compromising the quality and sensory characteristics of the final product.

## 1. Introduction

Food industrial waste represents a growing menace for modern society. The circularization of agro-industrial by-products is receiving increasing attention from food manufacturers, companies, and researchers. The main objective of good waste management practice is minimizing food waste by upcycling and creating a “zero waste society”.

Beer is a low-alcohol beverage, consumed worldwide, with a global production of about 1.91 billion hectoliters in 2019 [1]. Beer companies generate large amounts of waste in the form of spent grain, hot trub, and residual yeast. Usually, 100 kg of processed dry grain generates about 125–130 kg of wet spent grain with a moisture content of 80 to 85% obtained using filtration tanks, or 50 to 55% obtained in a press filter [2]. Hot trub is the second solid residue largely resulting from insoluble coagulation of mainly high-molecular-weight proteins.

Approximately 0.2 to 0.3 kg of hot trub (80–90% moisture) is formed for each hectoliter of beer [2]. This by-product mainly comprises high-molecular-weight protein, phenols, and lipids. Hop components with a lower solubilization may be precipitated by undergoing an electrostatic interaction with insoluble trub proteins. Currently, trub is valorized by mixing the brewer’s spent grains or other ingredients to prepare animal feed [3].

Trub could be exploited to formulate fortified foods owing to its higher nutritional profile. The raw material requires a debittering step before use as a food ingredient. Previous studies carried out a trub debittering method that enriches the by-product of protein and fiber fractions. Nevertheless, the protein contained in the resulting ingredient is very cross-linked and almost insoluble, particularly at low pH [4].

Nowadays, consumption of durum wheat pasta is increasing globally as it is an energy-dense meal. This staple food is rich in starch but poor in other nutrients. From this point of view, pasta represents a potential vehicle to deliver functional ingredients [5,6]. Accordingly, the literature reported positive results in terms of nutritional and technological aspects for pasta incorporated with functional components, including those obtained from agro-industrial by-products [5,6,7]. Furthermore, such innovative ingredients could modulate in vitro starch digestibility, thus improving the nutritional profile of a particular food product [8,9].

Few studies assessing the incorporation of debittered trub powder (DTP) in durum wheat pasta are available in the literature. Therefore, the present paper aims to evaluate the effects of DTP incorporation into fresh wheat-based pasta by investigating technological and sensorial properties. Lastly, we investigated in vitro starch and protein digestibility of DTP-fortified pasta samples.

## 2. Materials and Methods

### 2.1. Debittered Trub Powder Preparation

Fresh trub from pilsner-style beer production was supplied by the Mastro Matto brewery (Verona, Italy). In the initial step, excessive water was roughly removed from the hot trub by filtration using filter bags; the by-product was then stored at −18 °C.

For the debittering step, the trub and distilled water were mixed in a ratio of 25 g trub/L [4]. The slurry was then heated at 100 °C for 60 min; the sample was filtered as detailed above and the retained solid fraction was recovered. This procedure was repeated twice. The final solid fraction was named debittered trub (DT) and was dried in an oven (50 °C, 16 h).

Finally, the dried DT was milled and sieved to obtain debittered trub powder (DTP) with a particle size <200 µm. The DTP was stored in vacuum-sealed bags until use.

### 2.2. Preparation of Pasta with DTP

Durum wheat semolina (WS) was commercially purchased. The WS contained 69.50 g/100 g of total carbohydrates, 12 g/100 g of crude proteins, 1.70 g/100 g of total fats, and 2.50 g/100 g of total dietary fiber.

Fresh pasta was produced by replacing WS with 0, 5, 10 and 15 g/100 g of DTP to obtain PT0 (control), PT5, PT10 and PT15 samples, respectively. The dough was prepared by adding 35% *v*/*w* of tap water (37 °C) to WS or the different blends.

A professional pasta-making machine was used to produce fresh pasta samples, with extrusion through a 0.22 cm diameter bronze spaghetti die (Mod. Lillodue, Bottene, Marano Vicentino, Italy) and cutting the samples to 20 cm in length. Pasta samples were cooked in deionized water at a 1:10 (*w*/*v*) ratio.

### 2.3. Proximate Composition

The chemical compositions of trub, DTP and raw fresh pasta samples were determined according to AOAC standard methods: method 942.05 for ash, method 976.05 for crude protein, method 954.02 (without acid hydrolysis) for crude lipid, and method 996.11 for total starch. The total dietary fiber (TDF) contents were determined according to AOAC method 985.29 [10]. A Megazyme assay kit was used to assess free sugars using the K-SUFRG 06/14 (Megazyme, Wicklow, Ireland). The chemical compositions of trub and DTP were expressed as g/100 g of dry matter (DM), while the data from pasta samples were reported as g/100 g of wet matter (WM).

### 2.4. In Vitro Digestion of Pasta Samples for the Evaluation of Digestibility of Proteins

The protocol involved an oral, gastric, and intestinal phase with the modifications previously described by Rocchetti et al. [9].

Briefly, a 5.0 g sample of cooked pasta and a control (constituted by digestion fluids and enzymes) were put in nine tubes corresponding to the different times of digestion (salivary phase: S0; gastric phases: G0, G30, G60; intestinal phases: I0, I30, I60, I90, I120). The samples, in progressive succession, were hydrolyzed at 37 °C through (i) an oral phase, [5 mL of salivary fluid at pH 7.0 plus human salivary α-amylase (A1031; Sigma-Aldrich; Milan, Italy; 75 U/mL] for 2 min; (ii) a gastric phase [10 mL of a simulated gastric fluid at pH 3.0 plus pepsin (P7012; Sigma-Aldrich; 2.000 U/mL] for 60 min; and (iii) an intestinal phase [20 mL of simulated intestinal fluid at pH 7.0 plus pancreatin (P7545; Sigma-Aldrich; Milan, Italy; 100 U/mL] and bile salts (B8631; Sigma-Aldrich; Milan, Italy; 10 mM) for a further 120 min. At a specific time, the reactions were stopped with NaOH 1 M for the gastric phase and heat-shock treatment (100 °C, 5 min) for the intestinal phase, as described by Brodkorb et al. [11]. The samples were then centrifuged (6000 g, 10 min) and the supernatants were stored at −20 °C.

Primary Amino Nitrogen (PAN) groups were quantified using an Ortho-phthaldialdehyde (OPA)-based kit (PANOPA; Megazyme, Wicklow, Ireland). The results were calculated by subtracting the control value from the value of the samples.

### 2.5. In Vitro Starch Digestion of Pasta Samples

After the fully-cooked time had elapsed, pasta samples (100 mg) were processed as previously described by Rocchetti et al. [9].

Briefly, spaghetti samples (100 mg) were cooked for the fully-cooked time (see paragraph 2.6) and then minced in 4 mL of maleic buffer (pH 6) containing an enzyme mixture composed of amyloglucosidase (AMG; 4 mL; 300 U/mL; Megazyme, Wicklow, Ireland) and pancreatic α-amylase (40 mg; 3000 U/mg; Megazyme, Wicklow, Ireland). Samples were incubated in a shaking water bath at 37 °C. At 0, 30, 60, 120, and 180 min, the reactions were stopped with 8 mL of ethanol. Samples were then centrifuged at 2500 g for 10 min.

The amount of glucose was quantified by the glucose oxidase plus peroxidase (GOPOD)-based method (RESISTANT STARCH; Megazyme, Wicklow, Ireland).

### 2.6. Technological Properties

The moisture content, fully-cooked time (FCT), and the cooking loss (CL) of samples were determined by the AACC standard methods 44–15A, 66–51.01, and 66–50 [12].

The swelling index (SI) was determined as previously reported by Clearly and Brennan [13]. The SI is defined as the degree of water absorption in cooked pasta samples, due to protein hydration and starch gelatinization. A quantity of 20 g of pasta was cooked in 500 mL of deionized water at FCT. Subsequently, 2.5 g of cooked and minced pasta were placed in aluminum bowls at 105 °C for 24 h. After cooling in the dryer for 30 min, the SI was measured as shown below:$$\mathrm{SI}=\frac{\left(\mathrm{weight}\;\mathrm{of}\;\mathrm{cooked}\;\mathrm{pasta}-\;\mathrm{weight}\;\mathrm{of}\;\mathrm{pasta}\;\mathrm{after}\;\mathrm{drying}\right)}{\;\mathrm{weight}\;\mathrm{after}\;\mathrm{pasta}\;\mathrm{drying}}$$

Water activity (a_w_) was determined using a Hygropalm HC2-AWmeter (Rotronic Italia, Milano, Italy) at 23 °C [14].

### 2.7. Texture Analysis

Pasta samples were cooked in deionized water according their specific FCT and then dipped in cool deionized water for 45 s to stop the cooking process. Pasta firmness (AACC method 16–50; AACC 2020) [12] was measured with a light knife blade (A/LKB) and a speed of 0.17 mm·s^−1^. Adhesiveness was measured following the procedure detailed by Murray et al. [15]. For the analyses, a TA-XT2i Texture Analyzer (Stable Micro Systems, UK; 5 kg load cell) was used. Ten measurements were taken for each sample.

### 2.8. Color Analysis

A reflectance colorimeter (Illuminant D65) (Minolta Chroma meter CR-300, Osaka, Japan) was used to measure the surface color of pasta samples before and after cooking. The operation of the colorimeter is based on the color system CIE- *L*a*b**.

Analyses were performed at ten different points on each pasta sample [16].

### 2.9. Sensory Evaluation

A panel of 20 judges (10 men, 10 women; 22–59 years old) were trained to identify and give an intensity score, ranging from 1 for the lowest intensity to 9 for the highest intensity, to 12 sensory attributes (i.e., color uniformity and roughness for appearance, and pasta and malt aromas; adhesiveness, grittiness, gumminess, and elasticity for the texture, global flavor, sweetness, bitterness, and barley malt for the taste).

The pasta samples were submitted to the judges randomly. At the end of the test, we asked the judges to give an opinion on the overall acceptability of experimental samples using a nine-point hedonic scale (1 = dislike extremely; 9 = like extremely). Pasta samples were considered acceptable if their mean scores for overall acceptability were above 5 (neither like nor dislike) [14,16].

### 2.10. Statistical Analysis

The analyses were carried out at least in triplicate. All data represent the means of at least three measurements and results are expressed as the mean values ± standard deviation. The comparison of means was conducted using the analysis of variance (ANOVA) with a post hoc Tukey test at *p* < 0.05. Statistical analyses were performed by XLSTAT (Addinsoft SARL, Paris, France).

## 3. Results and Discussion

### 3.1. Proximate Analyses

Table 1 shows the chemical composition of trub, DTP, and the different pasta formulations. After the debittering process, the DTP carbohydrates numerically decreased because of the starch and sugar leaching into the boiling water, while the total dietary fiber (TDF) and the protein content numerically increased, reaching values of 59.5 g/100 g and 28.0 g/100 g, respectively. Similar results were reported elsewhere [4].

The experimental fresh pasta samples fell within the legal limits of a_w_ (0.92–0.97) and moisture content (>24%) [17]. In particular, the moisture content (g/100 g) was 27.25 ± 0.32 for PT0, 27.11 ± 0.32 for PT5, 29.46 ± 0.35 for PT10, and 28.04 ± 0.33 for PT15, while the a_w_ was 0.95 for all samples.

From a nutritional standpoint, all DTP-containing samples could benefit from the claim “high fiber content”, the dietary fiber content being >6 g/100 g. The PT5 and PT10 samples could be claimed as a “source of protein”, and PT15 as “high protein content” according to European Regulation (EC) n.1924/2006. The DTP inclusion in the pasta formulation increased the ash content (from 0.49 to 1.04 g/100 g WM for PT0 and PT15, respectively; *p* < 0.05) and slightly reduced the sugar content (from 2.0 to 1.9 g/100 g WM for PT0 and PT15, respectively; *p* < 0.05). Furthermore, the total starch underwent a progressive decrease with the increasing level of DTP in the pasta formulation (from 52.0 to 46.2 g/100 g WM for PT0 and PT15, respectively; *p* < 0.05). Taken together, the present findings appear in line with the chemical composition of DTP and with the inherent level of DTP in each pasta formulation. Similarly, Saraiva et al. reported that ash, protein, and fiber content increased, while carbohydrates decreased, in pasta fortified with an increasing level of trub in the recipe [18].

### 3.2. In Vitro Starch and Protein Digestion

In vitro starch digestibility was evaluated in the cooked pasta samples (Figure 1A). Overall, the trend indicates that the DTP inclusion reduced the extent of glucose release in a dose-dependent way only at the longest in vitro incubation time. Specifically, in the first 20 min of incubation, in vitro starch hydrolysis (expressed as g glucose/100 g DM) occurred quickly for all samples, reaching similar values (*p* > 0.05) at 20, 60, 120 min. In contrast, at 180 min of in vitro starch digestion, PT15 showed significantly (*p* < 0.05) lower values (i.e., 61.34 g glucose/100 g DM) than those of the control (i.e., 67.66 g glucose/100 g DM). The lower glucose release at 180 min of in vitro incubation could be partially due to starch content reduction in the DTP-containing samples. Moreover, a coating of DTP on the starch granules could have contributed to reducing the surface exposure to enzyme digestion by acting as a physical barrier. This was already reported in fortified spaghetti with soluble fiber and in pasta of different wheat types [19,20]. In addition, reduced starch gelatinization could occur when fibers compete with starch granules for water adsorption. The effect of gelatinization on in vitro starch digestibility was widely studied previously, and data from the literature indicate that a higher degree of starch gelatinization generally leads to greater starch digestibility, since the gelatinization can disrupt the inherent starch structure, thereby increasing the susceptibility of starch to enzyme hydrolysis [21]. Furthermore, it was widely reported that phenols may inhibit α-amylases and α-glucosidases, contributing to reducing in vitro starch hydrolysis [22]. However, further studies are necessary to confirm this hypothesis [22].

The in vitro protein-hydrolysis-simulated digestion is shown in Figure 1B. It should be noted that the total protein content in the experimental samples was higher than in the control. The protein content of the debittered trub utilized in the present study was lower than that described by Saraiva et al. [4]. Nevertheless, the protein concentration in hot trub depends on several factors and may vary between breweries. Moreover, the trub proteins and those of the DTP have specific chemical–physical characteristics. Firstly, their binding with tannins can reduce their digestibility [9]. In addition, trub proteins are not soluble, considering their origin and the debittering procedure, and are strongly cross-linked, and this could limit their accessibility by the digestive enzymes. From this point of view, the mere quantification of protein extrapolated by the nitrogen content (Nx6.25) in the products may not be representative of their bioavailability. For this reason, we studied the digestibility of the fortified product with an in vitro protocol [16]. We assumed that the measure of amino groups progressively released could express the bio-accessibility of protein to enteric enzymes. As expected, the number of PAN released during in vitro digestion does not seem proportional to the total nitrogen content of the experimental samples, excluding the final time of digestion (I120) of PT15. This effect suggests a partial resistance to the enzymatic digestion of debittered trub proteins. In any case, observing the progression of the release of amino groups during digestion, several points should be considered. During the simulated gastric phase, little proteolytic activity is present. On the contrary, an augment of solubilized amino groups is observable starting from I0. This result could not be due to immediate enzymatic activity but to a release from the matrices of peptides, insoluble at an acidic pH. Indeed, trub proteins have an isoelectric point of about 3.9 [4]. This increased solubility could facilitate protease hydrolysis. This phenomenon is of particular interest at the most extended times (I120) of digestion when 15%-fortified pasta showed a significant increase in the release of PAN compared with all the other samples.

### 3.3. Technological and Cooking Properties, and Texture of Pasta

The fully-cooked time (FCT) of pasta decreased in PT5 and PT10 (Table 2). This behavior was probably due to the fiber increase and to a gluten dilution effect that could accelerate water penetration in the pasta shape, thus allowing a faster starch gelatinization [5,23]. Instead, FCT increased in PT15, reaching a value higher than that of the control. Studies found an increased FCT for high-protein pasta fortified with different proteins and fiber-rich ingredients [24,25,26,27]. In contrast, Saraiva et al. reported different values and a progressive increase in FCT for dry pasta samples fortified with increasing levels (i.e., up to 10% *w*/*w*) of trub [18]. However, in our study, we focused on fresh pasta preparation. Moreover, for the DTP production, different treatment parameters were selected, including temperature, pH, drying condition, and the granulometry of the DTP. These differences may explain the disagreement between results from the different studies.

Cooking loss (CL) is an index to evaluate the capability of pasta of retaining its structural integrity during the cooking phase. The fortification of pasta with DTP increased the CL as a function of the level of DTP in the recipe. The PT15 sample was characterized by the highest CL, being 10.92% (*p* < 0.05). This value of solid loss is considered acceptable for fresh durum wheat pasta, being <12% [28]. The progressive increase in CL with the addition of DTP could be related to the levels of fiber in the sample. A high level of TDF could damage the gluten network, thus causing less solid retention in pasta and greater leaching in the cooking water [5]. Similar results were reported in pasta fortified with by-products such as brewers’ spent grain, grape pomace, olive pomace, onion skin, and fermented black chickpea [7,22,29,30,31,32].

The DTP inclusion level of 5 and 10 g/100 g in pasta formulation caused a significant reduction in the SI compared with that in the control (i.e., PT0). On the contrary, the SI increased for PT15, reaching similar (*p* > 0.05) values to those of PT0. A similar trend was previously reported for pasta fortified with olive pomace and kimchi fibers [7,33]. Authors indicated that the fiber content of the added ingredient could lead to the unravelling of the gluten structure, causing a greater exposure of starch to the boiling water, thus causing an increase in the SI values [7,33]. However, in the present study, this effect seems to be more complicated and not related to the increasing level of DTP in the recipe. This might be related to the effect of competitive hydration of fibers and starch, and to possible interactions of trub protein with the gluten network. This observation requires further investigation.

The substitution of WS with increasing levels of DTP in the fresh pasta formulation influenced the texture attributes of the samples (Table 2). In particular, firmness increased with increasing levels of DTP in the recipe, with the greatest value recorded for PT10 (i.e., 253.72 N; *p* < 0.05). An increase in pasta adhesiveness, ranging from −3.97 to −21.02 N for PT0 and PT10, respectively, was also reported (*p* < 0.05). A firmness-increasing effect was reported by Tolve et al. upon the addition of grape pomace in durum wheat pasta [22]. In contrast, a firmness-decreasing impact was measured in oat bran and inulin-enriched semolina pasta [34]. These discrepancies can be related to the source of the dietary fiber, the inherent level of inclusion, and the process of pasta making [23]. In addition, PT15 showed a specific behavior as FCT increased with the DTP inclusion only at the higher substitution level, showing a firmness decrement (i.e., 129.36 N) and adhesiveness values similar (*p* > 0.05) to those of PT0 (i.e., −7.60 N). Ribero et al. reported a similar behavior for pasta fortified with brown algae, suggesting that the fiber present in the ingredient can form a barrier to water uptake during cooking, delaying core pasta starch gelatinization [35]. This causes the pasta surface to be overcooked, while the starch core is still present, thus producing a softer texture and a longer FCT.

### 3.4. Color Analysis

The addition of DTP in pasta formulation influenced the color (Table 3). The brightness (*L** parameter) decreased progressively (*p* < 0.05) in both cooked and raw pasta, changing from PT0 to PT15. Differences between cooked and uncooked pasta were more evident for DTP-containing samples. Indeed, the cooking process darkened all the experimental samples, while the control showed no significant variation after the cooking step (*p* > 0.05). Nocente et al., Zarzycki et al., and Simonato et al. obtained similar results in pasta with brewer spent grain, flaxseed, and olive pomace, respectively [7,26,36]. The *a** parameter value increased slightly significantly (*p* < 0.05) for cooked DTP pasta compared with the uncooked counterparts. Fortification levels did not significantly influence the *a** values for either the uncooked samples among themselves, or the cooked samples among themselves. Indeed, DTP from beer by-products could contain significant quantities of chlorophyll from hops. Chlorophyll deterioration or leakage could explain this increment in the redness parameter [33]. The fortified samples *b** parameter values were significantly lower than those of the control pasta. In addition, raw uncooked fortified pasta had lower *b** values than those of the cooked samples for all levels of fortification. Saraiva et al. reported a similar trend for *L**, *b**, and *a** parameters in dry pasta formulated with increasing levels of debittered trub [18].

### 3.5. Sensory Evaluation

A quantitative descriptive analysis (QDA) was applied to observe possible changes in sensory characteristics determined by DTP fortification of pasta. In terms of appearance, PT0 and PT15 exhibited more color uniformity than PT5 and PT10 (Figure 2). In addition, all the fortified samples, especially PT5, were perceived as being rougher than the control. These results could be due to the different fully-cooked times required by the experimental samples. As for the aroma, pasta and malt odors showed a reverse behavior depending on the DTP inclusion level. The odor of pasta was perceived more strongly in the control sample (i.e., PT0) than in experimental samples where the odor of malt prevailed significantly. Regarding texture and tactile sensations, DTP fortification did not cause any statistical difference in adhesiveness, grittiness, gumminess, or elasticity (*p* < 0.05). However, slight grittiness increases in DTP-fortified samples were assessed by the panel, although these increases were not significant. On the contrary, elasticity was perceived to be higher in the control pasta than in the fortified samples. The addition of DTP did not have a significant impact on the overall flavor. However, bitterness and barley malt were perceived more in the fortified pasta samples, whereas sweetness decreased with the inclusion of DTP in the formulations.

The PT0 sample had the best overall acceptability by the panel with a score of 6.4, while PT15 had the lowest value (i.e., 3.8). PT5 and PT10 mean score values overcame the acceptability threshold of 5 (5.6 ± 1.5 and 5 ± 1.6, respectively).

### 3.6. Conclusions

A comprehensive characterization including technological, nutritional, and sensorial properties of fortified pasta was performed. All DTP-containing samples could benefit from the claim of “high fiber content”. Moreover, PT5 and PT10 samples could be claimed as a “source of protein”, and PT15 as “high protein content”. Even though the cooking loss increased with the addition of DTP to pasta formulations, all values were in the acceptability range for fresh pasta. The FCT, SI, and textural properties were peculiar in PT15, probably because of possible complex interactions between the gluten network and starch with DTP proteins and fibers, thus deserving further investigations. In the in vitro digestion, fortified samples showed a lower glucose release compared with that of the control at longer incubation times, while protein hydrolysis rose only in PT15. The QDA panel perceived higher malt aroma, grittiness, bitterness, and barley malt in the fortified samples. PT5 and PT10 samples overcame the sensory acceptability threshold of 5, while PT15 showed the lowest acceptability. In conclusion, DTP represents a suitable ingredient in fresh pasta formulation with an up to 10 g/100 g substitution level in the recipe to benefit from its nutritional composition without compromising the quality and sensory characteristics of the final product.

## Figures and Tables

**Figure 1 foods-11-02496-f001:**
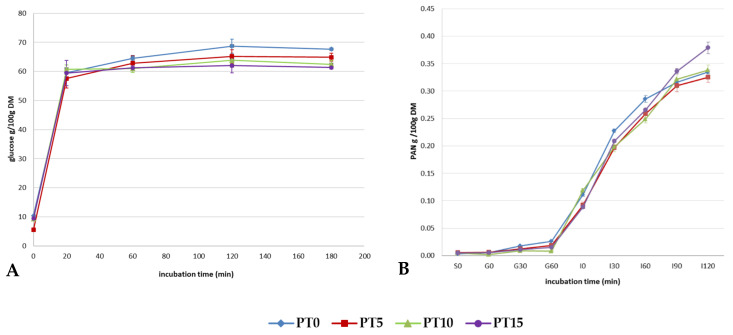
(**A**): graph of starch digestion of debittered trub powder (DTP) containing pasta vs. incubation time. (**B**): graph of protein digestion of DTP pasta vs. incubation time. PT0: pasta formulated with 100% durum wheat semolina. PT5, PT10, PT15: 95:5, 90:10, 85:15 durum wheat semolina:DTP, respectively.

**Figure 2 foods-11-02496-f002:**
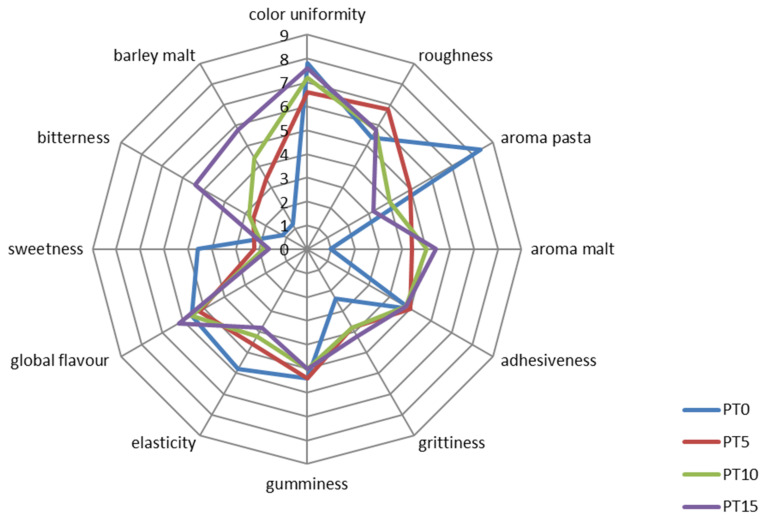
Sensory scores of quality attributes of pasta fortified with debittered trub powder (DTP) at different addition levels (PT0 blue line; PT5 red line; PT10 green line; PT15 purple line). PT0: pasta formulated with 100% durum wheat semolina. PT5, PT10, PT15: 95:5, 90:10, 85:15 durum wheat semolina:DTP, respectively.

**Table 1 foods-11-02496-t001:** Chemical composition of trub (T), debittered trub powder (DTP) (g/100 g dry matter) and wheat-based fresh pasta fortified with increasing levels of DTP in the formulation (g/100 g wet matter).

Samples	T	DTP	PT0	PT5	PT10	PT15
Ash	3.3 ± 0.01	4.2 ± 0.03	0.5 ± 0.01 ^a^	0.6 ± 0.01 ^b^	0.9 ± 0.01 ^c^	1.0 ± 0.03 ^d^
Fat	5.0 ± 0.1	6.1 ± 0.2	1.2 ± 0.1 ^a^	1.4 ± 0.1 ^b^	1.7 ± 0.1 ^c^	1.9 ± 0.11 ^d^
Sugar	17.3 ± 0.2	0.0	2.0 ± 0.0 ^a^	1.8 ± 0.0 ^b^	1.7 ± 0.1 ^c^	1.9 ± 0.1 ^b^
Protein	23.2 ± 0.70	28.0 ± 0.10	12.9 ± 0.3 ^a^	13.5 ± 0.06 ^b^	14.2 ± 0.2 ^c^	14.9 ± 0.17 ^d^
Total dietary fiber	26.9 ± 1.4	59.5 ± 1.6	3.4 ± 0.1 ^a^	11.0 ± 0.2 ^b^	11.8 ±0.1 ^b^	12.9 ± 0.2 ^c^
Total starch	22.9 ± 0.3	1.6 ± 0.10	52.0 ± 0.5 ^a^	49.8 ± 0.16 ^b^	47.0 ± 0.2 ^c^	46.2 ± 0.2 ^d^

Means in the same row with different superscripts differed at *p* < 0.05. PT0: pasta formulated with 100% durum wheat semolina. PT5, PT10, PT15: 95:5, 90:10, 85:15 durum wheat semolina:DTP, respectively.

**Table 2 foods-11-02496-t002:** Cooking quality parameters and texture analysis of fresh control pasta (PT0) and pasta samples fortified with different DTP substitution levels (5, 10, and 15 g/100 g: PT5, PT10, and PT15, respectively). Values are expressed as mean ± standard deviation.

Pasta Samples	Fully-Cooked Time (min)	Cooking Loss (%)	Swelling Index (g Water/g Dry Pasta)	Firmness (N)	Adhesiveness (N)
PT0	6.5	5.45 ± 0.17 ^a^	1.62 ± 0.06 ^a^	135.02 ± 14.53 ^a^	−3.97 ± 0,62 ^a^
PT5	3.75	6.16 ± 0.19 ^b^	1.17 ± 0.02 ^b^	216.47 ± 27.21 ^b^	−11.53 ± 1.57 ^b^
PT10	3	6.55 ± 0.12 ^c^	1.23 ± 0.007 ^b^	253.72 ± 21.87 ^c^	−21.02 ± 2.47 ^c^
PT15	6.75	10.92 ± 0.05 ^d^	1.71 ± 0.04 ^a^	129.36 ± 3.64 ^a^	−7.60 ± 0.28 ^a^

Values with different superscripts within the same column are significantly different for *p* < 0.05. PT0: pasta formulated with 100% durum wheat semolina. PT5, PT10, PT15: 95:5, 90:10, 85:15 durum wheat semolina:DTP, respectively.

**Table 3 foods-11-02496-t003:** Color analysis of uncooked and cooked control pasta (PT0) and pasta fortified with increasing levels of debitter trub powder (DTP) in the recipe (TP5, TP10, and TP15). The data, reported as mean ± standard deviation, are expressed as *L**, *a**, and *b** values.

Pasta Sample	Cooked/Uncooked	*L**	*a**	*b**
PT0	Uncooked	82.81 ± 1.69 ^a^	1.06 ± 0.56 ^c^	8.50 ± 2.97 ^a,b^
	Cooked	81.88 ± 0.75 ^a^	−1.21 ± 0.48 ^d^	10.25 ± 2.25 ^a^
PT5	Uncooked	68.47 ± 3.25 ^b^	3.94 ± 0.73 ^b^	−2.87 ± 0.91 ^e^
	Cooked	38.26 ± 3.02 ^e^	7.36 ±0.86 ^a^	6.86 ± 1.64 ^b,c^
PT10	Uncooked	62.48 ± 3.85 ^c^	3.98 ± 0.89 ^b^	−1.09 ± 1.25 ^e^
	Cooked	23.55 ± 3.67 ^f^	7.23 ± 1.01 ^a^	3.63 ± 1.24 ^c,d^
PT15	Uncooked	54.91 ± 5.08 ^d^	4.39 ± 0.32 ^b^	−1.42 ± 1.98 ^e^
	Cooked	23.43 ± 2.51 ^f^	7.93 ± 1.27 ^a^	4.86 ± 1.56 ^d^

Values with different superscripts within the same column are significantly different for *p* < 0.05. PT0: pasta formulated with 100% durum wheat semolina. PT5, PT10, PT15: 95:5, 90:10, 85:15 durum wheat semolina:DTP, respectively.

## Data Availability

Not applicable.

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
