# Peer review of "Durum Wheat Fresh Pasta Fortification with Trub, a Beer Industry By-Product"

_foods, 2022, doi:10.3390/foods11162496_

Round 1

Reviewer 1 Report

The article entitled “Durum wheat fresh pasta fortification with the trub, a beer industry by-product” is original. Using a food industrial waste on pasta formulation not only involves zero waste concept but also improve the nutritional profile a staple food that is globally consumed.

Minor revision should be made:

Line 196:  PT15 instead of TP15,  DTP instead of D>TP

Line 283:  Saraiva et al. (2019) instead of Saraiva et al (2019)

Reviewer 2 Report

Manuscript title: Durum wheat fresh pasta fortification with the trub, a beer in-2 dustry by-product

Manuscript reference: Foods 2022, 11, x. https://doi.org/10.3390/xxxxx

Specific comments to authors

1.      Certain corrections as well as suggestions have been incorporated in the PDF version of the manuscript. These shall be incorporated while revising.

2.       Page 3, line 121: Does this mean that only the residue was further reacted with gastric fluid or intestinal fluid as the case may be? Pl. make it clear.

3.       Page 3, line 125: Was there a control also because all the fluids also contain proteins? Pl. clarify.

4.       Page 3, lines 128-129: Why was the pepsin digestion step avoided? Usually in paste samples, a firm starch-protein network is formed which is partially responsible for the slow starch digestibility of pasta. Under gastric digestion, the protein  part gets hydrolysed and the starch is further acted upon by the intestinal enzymes. Without the pepsin step, will it be true simulation of the in vivo digestion? Please clarify.

5.       Table 1: In lines 71-73,it is mentioned that the DT was dried and the dry powder was stored till use. Then how could the values be expressed for the first two columns on a wet basis? Is it that the samples were again moistened? It is always better to compare on dry basis to take care of the differences in dry matter content. It is again mentioned under line 107 that dry matter was determined. Where is the data? If it is given, at least the readers could get a comparison of the moisture differences.

6.       Page 6, lines 245-246: Conveys a wrong meaning. The sentence should be recast as ' Fortification levels did not significantly influence the a* values either for the uncooked samples among themselves or the cooked samples among themselves'.

7.       Page 7, line 268: Usually under in vivo conditions as well as in in vitro studies,  a pepsin digestion step takes care of the removal of proteins from the starch-protein network in pasta. Unfortunately that step is missing in this study and done only for the protein digestibility study.

8.       Page 7, lines 275-278: Unnecessary correlation in this context. Could be deleted because no data are given on the phenol content of Trub.

9.       Page 8, line 321: addiction or addition?

Reviewer 3 Report

Comments for Author

I believe that the article entitled “Durum wheat fresh pasta fortification with the trub, a beer industry by-product” is an interesting one by it subject. The following changes is recommended to further improve the quality of manuscript.

Some remarks:

Abstract

-The abstract must be concise and only shows the major findings. Informative abstract is recommended for this manuscript. Each of the following parts might consist of 1–2 sentences. The parts include: background, aim or purpose, methods, findings/results and conclusion.

Introduction

The sentence “The trub could be exploited to formulate fortified foods thanks to its composition” change with “The trub could be exploited to formulate fortified foods owing to its higher nutritional profile”

Please use reference number instead of year i.e. Saraiva et al. (2019)

Cite the mentioned statement line 45-46 “Nevertheless, the protein contained in the resulting ingredient is very cross- 45 linked and almost insoluble, particularly at low pH.”

Line 59, sensory properties should be changed to sensorial properties

Material and methods

The swelling index (SI) line 88-89, The method should be explained, it is not sufficient in current form.

I strongly suggest that the basic analysis e.g. proximate composition should be written after the preparation of pasta and other should be sequenced in vitro studies, technological properties and sensory evaluation.

Results and discussion

The reasoning given for each result were inadequate. I recommend to rewrite the discussion section.

Why sugar content of DTP is 0.0????

The author stated that the increase in the total dietary fiber content of the samples was due to the debittering. Therefore, the reasons for the increase in fiber content should be explained with references.

Swelling index showed increasing with the addition of trub powder but only PT15 and control have higher values give a detail discussion regarding the current results. The given discussion is insufficient.

Conclusion

Remove the sentence “The present paper aimed to explore the use of DTP, a by-product generally rich in dietary fiber and protein, as a novel food ingredient in fresh pasta formulation.” Only present the finding in conclusion section.

The current manuscript showed plagiarism (32%). The below sources showed similarity index 10 and 6% respectively.

https://link.springer.com/article/10.1007/s11947-022-02838-9

https://publicatt.unicatt.it/handle/10807/154775

Round 2

Reviewer 3 Report

After modification according to the reviewers' comments, the manuscript can now be accepted for publication.